# Nigritoxin is a bacterial toxin for crustaceans and insects

Yannick Labreuche[1,2], Sabine Chenivesse[2], Alexandra Jeudy[2], Sophie Le Panse[3], Viviane Boulo[4], Dominique Ansquer[4], Sylvie Pagès[5], Alain Givaudan[5], Mirjam Czjzek[2] & Frédérique Le Roux[1,2]

The Tetraconata (Pancrustacea) concept proposes that insects are more closely related to aquatic crustaceans than to terrestrial centipedes or millipedes. The question therefore arises whether insects have kept crustacean-specific genetic traits that could be targeted by specific toxins. Here we show that a toxin (nigritoxin), originally identified in a bacterial pathogen of shrimp, is lethal for organisms within the Tetraconata and non-toxic to other animals. X-ray crystallography reveals that nigritoxin possesses a new protein fold of the α/β type. The nigritoxin N-terminal domain is essential for cellular translocation and likely encodes specificity for Tetraconata. Once internalized by eukaryotic cells, nigritoxin induces apoptotic cell death through structural features that are localized in the C-terminal domain of the protein. We propose that nigritoxin will be an effective means to identify a Tetraconata evolutionarily conserved pathway and speculate that nigritoxin holds promise as an insecticidal protein.

[1] Ifremer, Unité Physiologie Fonctionnelle des Organismes Marins, ZI de la Pointe du Diable, CS 10070, F-29280 Plouzané, France. [2] Sorbonne Universités, UPMC Paris 06, CNRS, UMR 8227, Integrative Biology of Marine Models, Station Biologique de Roscoff, CS 90074, F-29688 Roscoff cedex, France. [3] CNRS, FR 2424, Plateforme Merimage, Station Biologique de Roscoff, Place Georges Teissier, CS 90074, F-29688 Roscoff cedex, France. [4] Département Lagons, Ecosystèmes et Aquaculture Durables en Nouvelle-Calédonie, IFREMER, BP 2059, 98846 Nouméa cedex, New Caledonia. [5] UMR 1333 "Diversité, Génomes & Interactions Microorganismes - Insectes" Université Montpellier 2 - Place Eugène Bataillon, 34095 Montpellier cedex 5, France. Correspondence and requests for materials should be addressed to F.L.R. (email: frederique.le-roux@sb-roscoff.fr)

Arthropods are the most species-rich animal phylum on Earth. They represent more than 85% of the described animal species and are of tremendous importance to humans as food sources, pollinators, and producers of material goods (e.g., wax, honey, silk)[1–3]. However, some arthropods are also pests and vectors of disease and these interactions are likely to worsen due to range expansions caused by climate change and biological invasions[4–6]. A major challenge is to identify substances that are broadly effective and can be safely applied against known and emergent insect disease vectors and pests. A consensus reconstruction of arthropod relationships, based on molecular, morphological, and fossil data, has proposed that terrestrial insects (Hexapoda) are sister group to aquatic Crustacea and more distantly related to Chelicerates and Myriapods[1]. This suggests that insects and crustaceans (collectively termed the Tetraconata/Pancrustacea group[3, 7]) may share evolutionarily conserved pathways and toxins active against crustaceans may also be efficacious against insects, yet not harmful to more distantly related invertebrates and humans.

We previously identified a putative toxin locus specific to virulent strains of *Vibrio nigripulchritudo*[8, 9], a bacterial pathogen affecting shrimp in several areas in the Indo-Pacific[10, 11]. Expression of this gene by a non-virulent *V. nigripulchritudo* strain is sufficient to produce toxic culture supernatant, a characteristic of virulent strains[8]. Interestingly, a portion of this protein (thereafter named nigritoxin) shows significant sequence identity with antifeeding prophage 18 (Afp18), a partially characterized toxin found in the entomopathogenic bacteria *Serratia entomophila* and in the fish pathogen *Yersinia ruckeri*[12, 13], suggesting that nigritoxin may display toxicity against organisms beyond the known target, *Litopenaeus stylirostris* shrimp.

Here we describe the animal tropism and structure/function relationship of nigritoxin. Nigritoxin is sufficient to mimic the *V. nigripulchritudo* effect in vivo and has lethal activity against crustaceans and insects. X-ray crystallographic studies show that nigritoxin structure represents an entirely unique protein fold and indicate that Afp18 and nigritoxin do not possess the same

biological functions. Nigritoxin is comprised of three domains, with the N-terminal domain being essential for cellular translocation and likely encoding specificity for Tetraconata, and the C-terminal domain being involved in cell death induction through apoptotic mechanisms. We then show that nigritoxin is a deadly toxin that targets a Tetraconata evolutionarily conserved pathway.

## Results

**Nigritoxin mimics *Vibrio* effect on shrimp hemocytes.** To fulfill the criteria of a bacterial toxin, administration of the purified component at low dose to the animal should elicit key symptoms observed in the host infected with the pathogen. Recombinant purified nigritoxin was intramuscularly injected into *L. stylirostris* shrimp and we found that 3 ng g$^{-1}$ body weight (40 fmol g$^{-1}$) of protein was sufficient to kill 50% of the tested animals (LD$_{50}$) within 5–10 h (Fig. 1a; Supplementary Fig. 1). In *L. stylirostris*, infection with *V. nigripulchritudo* results in septicemia[10] and the only histopathological sign associated to infection is abnormal nuclei in the circulating hemocytes (Supplementary Fig. 2). We thus compared the cytopathogenic effects induced in shrimp hemocytes after injection of either the nigritoxin or *V. nigripulchritudo* by transmission electron microscopy (TEM) (Supplementary Fig. 3). In control animals (shrimp injected with saline or a non-virulent *V. nigripulchritudo* strain), the cells contained normal and intact nuclei with heterochromatin visible in peripheral patches and as a central mass (Supplementary Fig. 3a, d). In shrimp injected by the virulent strain, severe alterations were observed in the hemocytes including membrane disruption, chromatin condensation (pyknosis), and nucleus fragmentation (karyorrhexis), a characteristic of cell undergoing necrosis or apoptosis (Supplementary Fig. 3c). Similar alterations in nuclei were observed in the hemocytes sampled from nigritoxin-injected shrimp (Supplementary Fig. 3b), although more severe cell damage (cell lysis and vacuolization), that correlates to a faster occurrence of mortality, was also observed.

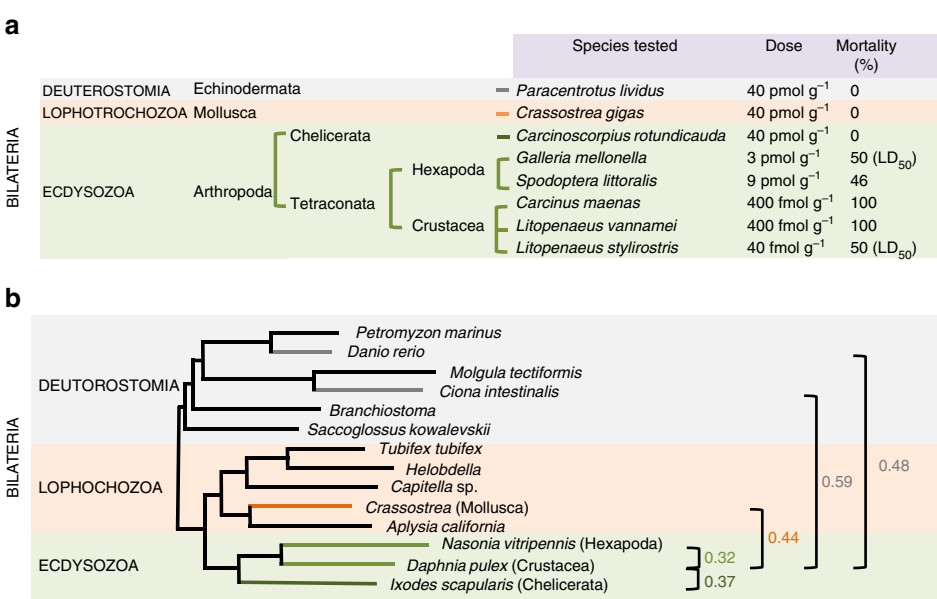

**Fig. 1** Toxicity of the nigritoxin for several species of bilateria. **a** Percentage of mortality obtained after injection of the indicated dose of nigritoxin to different species representative of diverse phyla within the Bilateria. The LD$_{50}$ was determined for *Litopenaeus stylirostris* and *Galleria mellonella* using Probit analysis (Supplementary Fig. 1). **b** Evolutionary distances in substitutions/site (right side) between Crustacean and Hexapoda (light green), Chelicerata (dark green), Mollusca (orange), and two Deutorostomia *Ciona intestinalis* and *Danio rerio* (gray). The tree is adapted from Nosenko et al.[52] who analyzed a matrix composed of 22,975 homologous amino-acid positions

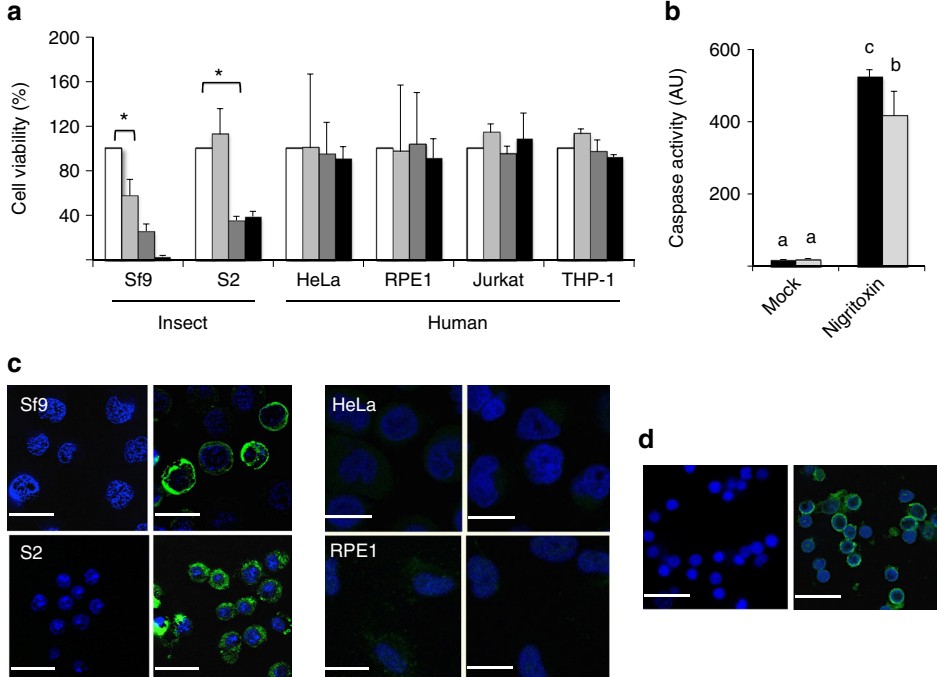

**Fig. 2** Cytotoxic activity of the nigritoxin on different cell lines. **a** Two insect (Sf9 and S2) and four human (HeLa, hTERT RPE-1, Jurkat, and THP-1) cell lines were incubated with nigritoxin (1.2 μM) for 0, 6, 12, and 24 h (white, light gray, dark gray, and black bars, respectively). The cytotoxicity expressed in percentage (y-axis) was monitored using Alamar Blue assay. The experiment was performed twice in triplicate and data are presented as mean ± S.D. A star indicates significant difference between treatment (ANOVA, $p < 0.05$) **b** Sf9 and S2 cells (black and gray bars, respectively) were either mock-treated or incubated with 1.2 μM nigritoxin for 12 h. Caspase activity was determined using Ac-DEVD-MCA as a substrate. The experiment was performed in triplicate and data are presented as mean ± S.D. Means with the same letter are not significantly different from each other (Kruskal–Wallis, $p < 0.05$). **c** Insect and human cell lines were incubated with nigritoxin (1.2 μM) for 6 h, then washed, fixed, and reacted with a specific anti-nigritoxin polyclonal antibody and Alexa Fluor-488-conjugated secondary antibody (green). Nuclei were stained with DAPI. Left image, mock-treated cells; right image, nigritoxin-treated cells. **d** Primary hemocytes from shrimp *Litopenaeus vannamei* were treated and analyzed as described in **c**. Scale bars: 20 μM

**Nigritoxin is lethal for Tetraconata**. We explored whether nigritoxin has toxicity against organisms beyond the known target, *L. stylirostris*. Recombinant nigritoxin or a vehicle control was injected into representatives from diverse invertebrate groups (Fig. 1). Control injections did not result in lethality or observable phenotypes in any tested animals. All crustaceans died within 24 h following injection of 30 ng g$^{-1}$ (400 fmol g$^{-1}$) body weight of protein (Fig. 1a), a dose comparable to that of venom toxins with activity on marine crustaceans[14, 15]. Protein was also injected into the hemocoel of larvae from the lepidoptera *Galleria mellonella* and *Spodoptera littoralis*. Nigritoxin was found to be lethal to both insects (Fig. 1a), with an LD$_{50}$ of 270 ng g$^{-1}$ (3 pmol g$^{-1}$) for *G. mellonella* (Supplementary Fig. 1) and LD$_{46}$ of 788 ng g$^{-1}$ (9 pmol g$^{-1}$) for *S. littoralis* (Fig. 1a). On the other hand, nigritoxin did not cause larval mortality when ingested (Supplementary Table 1), although some bacterial toxins have been shown to exert oral toxic capability in lepidoptera[16]. Finally more evolutionary distantly related organisms, i.e., chelicerates, mollusks, and echinoderms (Fig. 1b) were still alive 1 week after injection of up to 3.2 μg g$^{-1}$ (40 pmol g$^{-1}$) nigritoxin (Fig. 1a).

**Nigritoxin induces apoptotic cell death in insect cells**. Further investigation of nigritoxin cytotoxicity was performed using insect and human cell lines. Treatment of the insect cell lines Sf9 and S2 with nigritoxin induced a dose-dependent decline in cell viability, although S2 cells were less sensitive (Supplementary Fig. 4). A dose of 1.2 and 12 nM nigritoxin resulted in significant cell death within 12 h for Sf9 and S2 cells, respectively. In contrast, up to 1.2 μM nigritoxin did not kill a variety of human cell lines (Fig. 2a) nor have an effect on cell cycle progression

(Supplementary Table 2). In insect cells, nigritoxin treatment was associated with cell shrinkage, blebbing, vacuolization, and DNA fragmentation and condensation (Supplementary Fig. 5), suggesting the occurrence of apoptotic cell death. In support of this idea, we observed that treatment of insect cells with the broad-spectrum caspase inhibitor, zVAD-fmk, which is known to prevent apoptotic cell death[17], repressed the cytotoxic effect induced by nigritoxin (Supplementary Fig. 6). Furthermore, nigritoxin treatment of cells resulted in markedly increased caspase activity, as typically observed following induction of apoptosis (Fig. 2b). Caspase induction, like cell death, was less marked in S2 cells than in Sf9 cells treated with nigritoxin. Interestingly, S2 also appeared less sensitive to staurosporine, a well-known initiator of apoptosis (Supplementary Fig. 7). Immunofluorescence analyses show that while two types of cultured insect cells are susceptible to nigritoxin, human cells may be refractory to the toxin due to the inability of the toxin to bind and/or be internalized (Fig. 2c). Cellular translocation of the protein was also observed when shrimp primary hemocytes were incubated with nigritoxin in vitro (Fig. 2d). Finally, time course experiments show that nigritoxin increasingly accumulates in the cytosol of intoxicated Sf9 cells (Supplementary Fig. 8a) with no apparent protein processing (Supplementary Fig. 8b).

**Nigritoxin structure represents a unique protein fold**. The crystal structure of the nigritoxin was solved by the single anomalous diffusion (SAD) method, using a selenomethionine-labeled protein and refined at 2.1 Å resolution. The overall structural topology of nigritoxin is divided into three globular domains: a N-terminal (aa 17–268), a central (aa 276–460), and

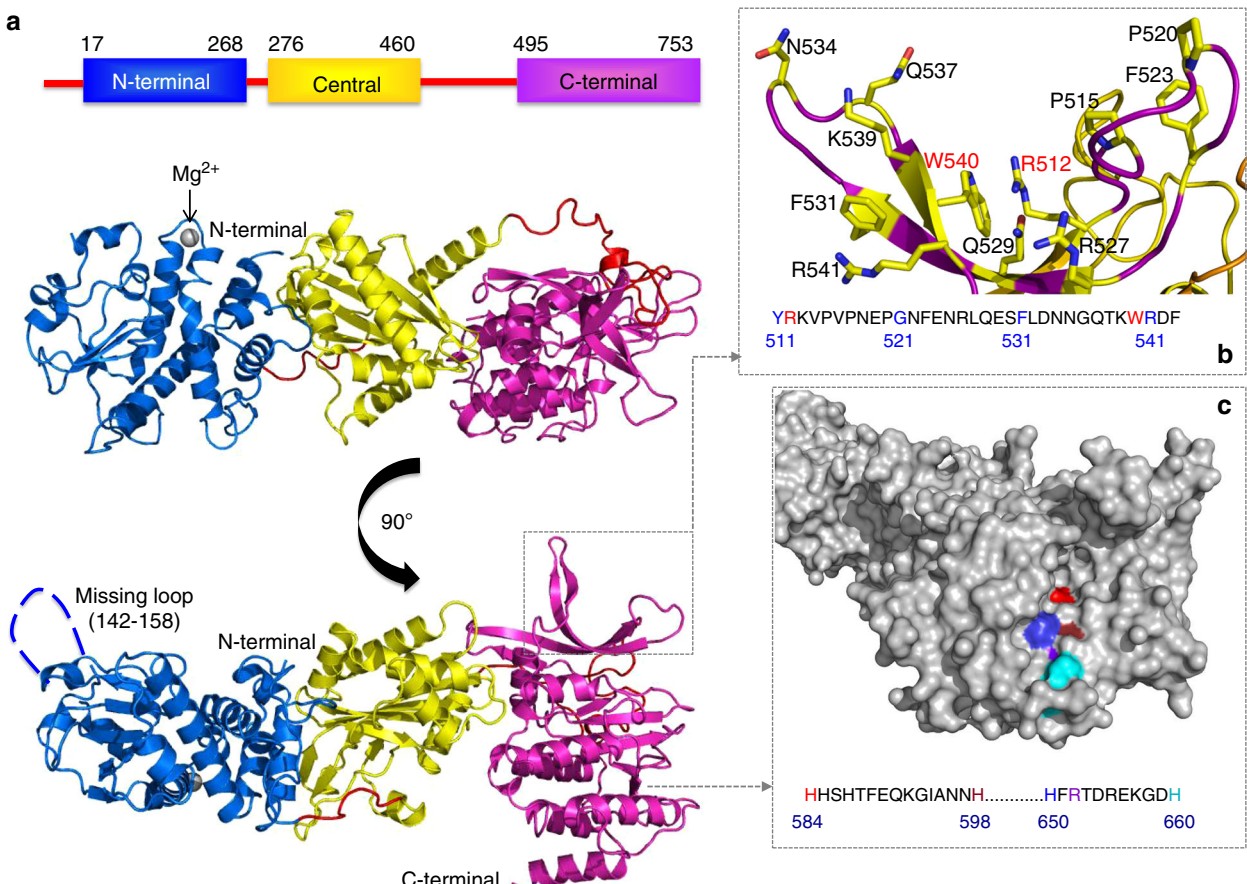

**Fig. 3** Ribbon representation of the overall 3D structure of the nigritoxin molecule. The polypeptide chain is color-coded following the domain architecture (blue: N-terminal; yellow: central; magenta: C-terminal; domains) schematically represented in the upper panel of **a**. The central panel highlights the 3D domain organization observed in the crystal structure, revealing the $Mg^{2+}$ bound to a surface loop of the N-terminal domain and the linker regions (orange) (see also Fig. 4). The lower panel displays the 3D structure turned by 90° around the horizontal axis, showing a disordered loop in the N-terminal domain for which no electron density is visible in the crystal structure. The overall fold as well as those of the three individual domains do not superimpose to any known structure when submitting a "blast against PDB". **b** Close-up view of a particular structural feature present in the C-terminal domain, which is formed by two loops projecting into the solvent region. The arginine (R512) and tryptophan (W540) residues that stack against each other, as well as some selected residues pointing toward the exterior of the protein are highlighted in stick representation. **c** Surface representation of the C-terminal domain (gray), highlighting a local groove, in which a series of histidine and arginine residues (surface color corresponds to the color of the residue in the sequence extract) form a large patch that could represent a region important for the biological function. The patch is localized in front of a cavity (Supplementary Fig. 12)

C-terminal domain (aa 495–753) (Fig. 3a; Supplementary Fig. 9). A surface loop in the N-terminal domain contains a metal ion that was modeled as $Mg^{2+}$, since this ion is present in the crystallization conditions (Fig. 4). However, treatment of Sf9 cells with the chelating agent EDTA did not substantially alter the cytotoxic effect induced by nigritoxin (Supplementary Fig. 10), suggesting that the role of $Mg^{2+}$ coordination is structural and not important for the activity. The C-terminal domain of nigritoxin contains two interesting and not yet described structural features: residues 512–544 form two large loops, each composed of two antiparallel strands that project out of the globular surface like "rabbit ears" (Fig. 3b; Supplementary Fig. 11). The bases of these protruding loops appear to be stabilized through a hydrophobic stacking of the side chain of Arg512 with the side chain of Trp540. The second feature is located on the concave side of the second, larger β-sheet, where four histidine residues and one arginine are concentrated in a groove with maximal distance of 10 Å (Fig. 3c; Supplementary Fig. 12) creating a large and positively charged patch at the protein surface that has yet to have a defined biological function. Comparative analyses with Blast against PDB, DALI, Phyre2, or ProFunc revealed that neither the

structure of full-length nigritoxin nor that of its individual subdomains resembles any known fold or protein structure, meaning that to date this protein structure is an "orphan", making the identification of an active site difficult. Consequently, the entire nigritoxin protein can be considered to possess a new protein fold, and following the classification of CATH (http://www.cathdb.info/) belongs to the α/β class. According to the statistics at the RCSB Protein Data Bank (www.rcsb.org/pdb/statistics/), new folds have not been reported or deposited in the last 4 years.

**Nigritoxin structural domains have distinct roles**. Site directly mutated and/or truncated forms of nigritoxin were used to define functions for its constituent regions. Double substitution with alanine of the two amino acids (Arg512 and Trp540) suspected to be involved in the stabilization of the protruding loops did not affect toxicity to Sf9 cells (Fig. 5a). On the other hand, deletion of the protruding loops eliminated toxicity, as did replacement with alanine of two histidine residues (His598 and His650) that are suspected of contributing to a functional "hotspot" (Fig. 5a). None of these mutations prevented protein internalization

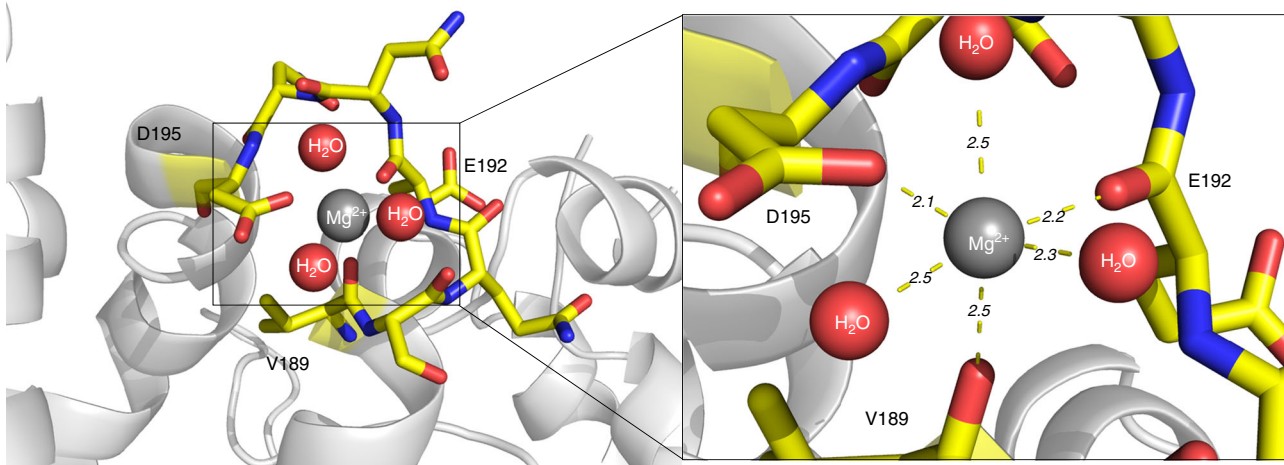

**Fig. 4** Coordination sphere of the magnesium ion. The residues of the loop carrying the $Mg^{2+}$ binding site are represented as sticks. The $Mg^{2+}$ ion is coordinated by the main chain carboxyl groups of V189 and E192, and the side chain carboxyl group of D195. The octahedral coordination sphere is completed by three water molecules. The right panel shows a close-up view of the binding site, indicating the distances to the coordinating atoms in Å

(Fig. 5b) and altered the folding of the proteins as measured by dynamic light scattering. These data suggest that toxicity is mediated at least in part through nigritoxin's C-terminal domain, as does the fact that no toxicity was induced by constructs lacking this domain. However, the C-terminal domain was not sufficient to induce Sf9 cell death (Fig. 5a). Immunofluorescence assays suggest that this may reflect a role for nigritoxin's N-terminal domain in toxin internalization, as only polypeptides that included the N-terminal domain showed marked intracellular accumulation (Fig. 5b). The truncated forms of nigritoxin were similarly detected by sodium dodecyl sulfate (SDS)–polyacrylamide gel electrophoresis (PAGE) and Western blot with polyclonal anti-nigritoxin antibodies, showing that these results were not due to protein instability or lack of detection by the antibodies (Supplementary Fig. 13). Thus, our working hypothesis is that the toxin's structural domains have distinct roles: the N-terminal domain mediates toxin internalization by target cells, while the C-terminal domain is necessary to induce cell death through apoptosis.

**Nigritoxin and Afp18 act by distinct mechanisms.** A portion of nigritoxin shows substantial sequence identity (≈28%) with Afp18, the toxin component of the phage tail-derived protein translocation system identified in several bacterial pathogens[12, 13]. Similarity is limited to nigritoxin's N-terminal domain (aa 17–228) and part of the middle domain (aa 276–373), and it corresponds to only a fragment of Afp18 (Supplementary Fig. 14). Highly conserved amino acids are largely localized within the hydrophobic core of the nigritoxin structure, suggesting that they may contribute more to maintenance of fold structure than to specific toxin activity (Supplementary Figs. 15, 16). Notably, Afp18 lacks homology to the region associated with nigritoxin toxicity, and nigritoxin likewise lacks homology to Afp18's C-terminal domain, which encodes a glycosyltransferase associated with inhibition of RhoA activation in *Y. ruckeri*[13]. Collectively, these data suggest that nigritoxin and Afp18 toxicity are produced through distinct mechanisms. Toxin internalization processes also appear to differ: in contrast to nigritoxin, which can enter host cells, Afp18 internalization is mediated by a phage tail-like particle with structural similarity to R-type pyocins[12]. Genes encoding such structures could not be identified in the *V. nigripulchritudo* genomes[8]. Thus, although Afp18 and nigritoxin may have a common evolutionary origin, subsequent domain shuffling

and divergent evolution appear to have resulted in distinct roles for these proteins.

## Discussion

Combining experimental infection, cellular, and structural biology data, we demonstrated that nigritoxin is a potent toxin with lethal effect for crustaceans and insects. Our results suggest that nigritoxin targets a pathway that has been evolutionarily conserved between Crustacea and Hexapoda, in line with the Tetraconata (Pancrustacea) hypothesis for arthropod phylogeny. Although it is necessary to confirm these data by testing more animal species in experimental infection, further exploration of the in vitro cytotoxicity of nigritoxin confirmed that a crustacean-specific receptor and/or means of translocation is conserved in insect cells and targeted by the N-terminal domain of nigritoxin, which is required for its internalization.

Nigritoxin not only possesses a new protein fold but also seems to act by an original mechanism leading to apoptotic cell death[18]. Nigritoxin is secreted by *V. nigripulchritudo* through an as yet unknown process[8]. As the purified protein induces toxicity, nigritoxin acts as a homoprotein and does not require a secretion system to be delivered into the eukaryotic host cell[19] nor need to be transported by outer membrane vesicles[20]. Nigritoxin does not disrupt membrane integrity by forming pores[21]. The target eukaryotic cells rapidly internalize nigritoxin, which then accumulates in the cytoplasm with no apparent protein degradation or processing. This suggests an intracellular mode of action of the nigritoxin. Consequently nigritoxin may not act as a host receptor agonist or antagonist, triggering signal transduction pathways[22], although accumulating evidences revealed that the endocytosis of receptor–ligand contributes to generate cellular signals[23, 24]. Finally although nigritoxin causes chromatin damage, it is not translocated into the nuclear compartment, such as cytolethal distending toxins[25]. The primary target of nigritoxin may within the cytoplasm and could involve caspases, the cellular executioners of apoptosis[26]. The C-terminal domain and more particularly the "rabbit ears" and functional "hotspot" are essential for nigritoxin cytoplasmic activity although a role of the two other domains cannot be excluded based on the present data. Hence several relevant questions need to be further addressed, such as: (i) does internalization implicate a surface receptor(s) and/or lipid raft? (ii) does internalization occur via endocytosis? (iii) what is the route of the toxin within the cytoplasm (intracellular

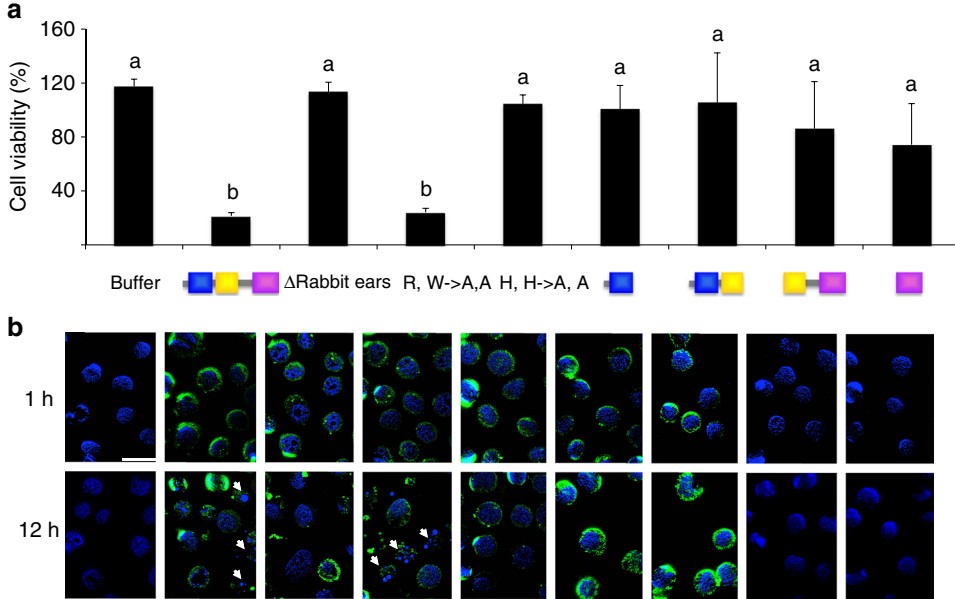

**Fig. 5** Functional role of nigritoxin domains. **a** Cytotoxic activity of wild type (domains schematically represented in blue: N-terminal; yellow: central; magenta: C-terminal), truncated forms, and mutants (Δrabbit ears: deletion of the protruding loops; R,W->A,A: Arg512 and Trp540 replaced by two Ala; H, H->A, A: His598 and His650 replaced by two Ala) of nigritoxin on insect cells. Sf9 cells were incubated with 1.2 μM of truncated forms or mutants of nigritoxin or with the same volume of protein suspension buffer for 12 h. The cytotoxicity was monitored using Alamar Blue assay. For each condition, the cell viability was compared to mock (untreated) cells set at 100% viability and expressed in percentage (y-axis). The experiment was performed twice in triplicate and data are presented as mean ± S.D. Means with the same letter are not significantly different from each other (Kruskal–Wallis, $p < 0.05$). **b** Cellular localization of wild type, truncated, and mutant versions of nigritoxin in Sf9 cells. Cells were incubated with purified proteins/fragments (1.2 μM) for 1 and 12 h, fixed and detected by immunochemistry. Nuclei are stained in blue with DAPI. White arrows indicate nuclei damages. Scale bars: 20 μM

trafficking, target organelles such as mitochondria, target molecules)?

Typical signs of apoptosis were also observed in hemocytes of nigritoxin-injected shrimp and mimic cytopathological symptoms observed in animal infected by *V. nigripulchritudo*. This suggests that nigritoxin is a major virulence factor of *V. nigripulchritudo* and the hemocyte its primary target. In support of this assumption, the nigritoxin was shown to be only active when delivered into the animal circulatory system, unlike some bacterial toxins displaying oral toxicity, such as the 3D-Cry toxins, which interact with specific receptors located on the insect gut epithelium leading to pore formation[27, 28] or the PirAB[vp] toxin in shrimp[29]. This may be explained either by the failure of the toxin to cross the peritrophic barrier, a chitin-rich matrix covering the digestive tube of insects and crustaceans, or more plausibly by the fact that gut cells do not represent the primary target of the nigritoxin.

The nigritoxin LD$_{50}$ appears to be of the same order of magnitude as that of other toxins tested in marine invertebrates[14, 15] or in insects[30, 31]. Targeting a broad range of insect species represents an alternative to the narrow activity spectrum of commercialized biological insecticides such as *Bacillus thuringiensis* toxins (Bt)[28], which are not toxic to some key insect pests, including the hemipteran (e.g., sap-sucking insects) pests[32]. We speculate that nigritoxin holds promise as an insecticidal protein but delivery systems have to be devised to access its target site, i.e., the animal circulatory system.

## Methods

**Bacterial strains.** The *V. nigripulchritudo* strain SFn1 (virulent) and VN110 plasmid cured derivative (non-virulent)[9] were grown in marine broth or on marine agar (Difco) at 30 °C. *Escherichia coli* strain TOP10 (Invitrogen) was used as plasmid host for cloning. *E. coli* strains BL21 (DE3) and B834 (DE3) (Novagen) were used for protein and selenomethionine-substituted protein production, respectively (Supplementary Table 3). Strains were grown at 37 °C in Luria–Bertani (LB) broth or agar (Difco) with 100 μg ml$^{-1}$ of carbenicillin, or in ZYP 5052 (for 1 l:

928 ml ZY (10 g l$^{-1}$ N-Z-amine AS, 5 g l$^{-1}$ yeast extract); 1 ml 1 M MgSO4; 1 ml 1000× metal mix (50 mM FeCl₃, 20 mM CaCl₂, 10 mM each of MnCl₂ and ZnSO₄, and 2 mM each of CoCl₂, CuCl₂, NiCl₂, Na₂MoO₄, Na₂SeO₃, and H₃BO₃ in ~50 mM HCl); 20 ml 50 × 5052 (25% glycerol, 2.5% glucose, and 10% α-lactose); 50 ml 20× NPS (0.5 M (NH₄)₂SO₄; 1 M KH₂PO₄; 1 M Na₂HPO₄); 1 ml carbenicillin 50 mg ml$^{-1}$) or PASM 5052[33] (for 1 l: 721 ml sterile water; 0.8 ml 1 M MgSO₄; 0.8 ml 1000× metal mix; 16 ml 50 × 5052; 40 ml 20× NPS; 100 nM vitamin B₁₂; 200 μg ml$^{-1}$ of each of 17 amino acids (no C, Y, and M); 10 μg ml$^{-1}$ methionine; 125 μg ml$^{-1}$ Se–Met; 1 ml carbenicillin 50 mg ml$^{-1}$) medium for induction of protein expression in BL21 (DE3) or B834 (DE3) strains, respectively.

**Molecular biology.** The nigritoxin gene (CDS) was PCR amplified using primer pair 090413-2/090413-3 (Supplementary Table 4), strain SFn1 DNA as target and Herculase II fusion proof reading polymerase according to the manufacturer instructions (Agilent). The amplicon was digested by *Bam*HI and *Xho*I and cloned into the pFO4 plasmid[34] (modified from pET15b, Novagen, USA) that contains an histidine tag at the N-terminal for affinity purification. Nigritoxin N-terminal domain (aa 1–270), N-terminal, and central domain (aa 1–457) and C-terminal domain (aa 486–757) were generated following the same procedure using primer pairs 090413-2/290914-2, 090413-2/290914-3, and 290914-1/090413-3, respectively. Cloning of the central and C-terminal domain (aa 271–757) into the pFO4 plasmid was performed using the Gibson assembly method according to the manufacturer's instructions (New England Biolabs, NEB) using primer pairs 200315-3/200315-5 and 200315-1/200315-2 to amplify the nigritoxin-targeted region and pFO4, respectively. Mutagenesis was performed by Gibson assembly using primers 060515-2/-3 (Arg₅₁₂ to Ala), 060515-4/-5 (Trp₅₄₀ to Ala), 310816-3/-4 (His₅₉₈ to Ala), 310816-5/-6 (His₆₅₀ to Ala), 310816-11/-12 (Thr₅₁₀ to Arg₅₄₁ to GGG), and primers (090413-2/090413-3) spanning the gene. All constructs were confirmed by sequencing prior to BL21(DE3) transformation.

**Purification of recombinant proteins.** After expression in 200 ml ZYP 5052 at 20 °C for 3 days, recombinant proteins were purified by resuspending cells in 4 ml suspension buffer (50 mM Tris pH 8.0, 25% sucrose, 100 μg ml$^{-1}$ lysozyme) and incubating 10 min at RT[35]. Next 8 ml of lysis solution (20 mM Tris pH 7.5, 100 mM NaCl, 1% Triton X-100, 1% deoxycholate) supplemented with antiprotease (complete, EDTA-free, Roche) was added and incubated stirring at 4 °C for 10 min. Finally, MgCl₂ and DNase were added at 5 mM and 6 U ml$^{-1}$, respectively, and the lysate was incubated for 20 min at RT prior to 45 min centrifugation at 13,865×g and 4 °C. The 0.2 μm-filtered supernatant was run on a 5 ml HisTrap HP column (GE healthcare Life Science) previously equilibrated with buffer A (20 mM NaPhosphate pH 7.4, 290 mM NaCl, 5 mM imidazole), then washed with buffer A

**Table 1 Data collection and refinement statistics for the crystal structures of the Se–Met-labeled and native nigritoxin molecules**

|  | Nigritoxin_Semet | Nigritoxin_native |
|---|---|---|
| **Data collection** | | |
| Beamline | PROXIMA 2 | ID29 |
| Wavelength(s) | 0.9793 | 1.0332 |
| Space group | I 23 | I 23 |
| Unit cell | $a = b = c = 183.32$ Å | $a = b = c = 185.16$ Å |
|  | $\alpha = \beta = \gamma = 90.00°$ | $\alpha = \beta = \gamma = 90.00°$ |
| Resolution range[a] (Å) | 45.83–3.02 | 65.46–2.10 |
|  | (3.20–3.02) | (2.21–2.10) |
| Total data | 449,101 | 205,825 |
| Unique data | 39,002 | 60,874 |
| Completeness (%) | 99.9 (99.8) | 99.3 (97.7) |
| Mean $I/\sigma$ $(I)$ | 15.13 (1.90) | 13.3 (3.3) |
| $R_{sym}$[b]; $R_{pim}$[c] (%) | 0.22 (1.26); 0.05 (0.20) | 0.06 (0.39); 0.03 (0.21) |
| Redundancy | 11.5 | 3.4 |
| Anom CC | 54 | – |
| Anom FOM | 0.353 | – |
| **Refinement statistics** | | |
| Resolution range | | 70.00–2.1 (2.155–2.10) |
| Unique reflexions | | 57,805 (4120) |
| Reflexions $R_{free}$ | | 3069 (234) |
| $R/R_{free}$ (%) | | 22.9/26.5 (25.3/29.6) |
| RMSD bond lengths | | 0.019 Å |
| RMSD bond angles | | 1.95° |
| Overall B factor (Å$^2$) | | 48.78 |
| B factor: molecule A (Å$^2$) | | 40.74 |
| B factor: solvent (Å$^2$) | | 57.67 |

[a]Values in parentheses concern the high-resolution shell
[b]$R_{sym} = \Sigma|I - I_{av}|/\Sigma|I|$, where the summation is over all symmetry equivalent reflections
[c]$R_{pim}$ corresponds to the multiplicity weighted $R_{sym}$

and eluted using a gradient of 1–100% buffer B (20 mM NaPhosphate pH 7.4, 290 mM NaCl, 0.5 M imidazole). The protein was concentrated using an Amicon stirred cell concentrator (molecular weight cutoff 10 kDa), passed through a Superdex S200 column (GE healthcare Life Science), eluted in Hepes 20 mM NaCl 150 mM pH 7.4, and finally concentrated as described above. Proteins were assessed as having >95% purity by SDS–PAGE. The concentration was calculated from the $A_{280}$ and the extinction coefficient calculated using the ProtParam tool from ExPASy ($\varepsilon_{0.1\%}$ of 0.69). For 3D structure resolution, the seleno-L-methionine (Se–Met) labeling procedure was performed by growing B834 (DE3) cells in 1 l of PASM 5052 medium, using the same purification method except that the final buffer contained 5 mM of tris (2-carboxyethyl) phosphin. The recombinant proteins used for structural determination and functional assays retained the histidine tag without impacting the structure and function of the protein.

**Crystallization and crystal structure determination.** Single crystals of native and Se–Met-labeled nigritoxin (10 mg ml$^{-1}$) were grown by the hanging drop method over a reservoir containing 100 mM Bis-Tris propane buffer at pH 8.5, 24% (w/v) PEG 3350, 0.1 M sodium fluorate, and 100 mM Bis-Tris propane buffer at pH 8.5, 20% (w/v) PEG 3350, 0.1 M sodium fluorate. To collect data at cryogenic temperature, crystals were rapidly soaked in a cryo buffer that was identical to the reservoir solution supplemented with 10% glycerol and frozen in a stream of nitrogen at 100 K. Nigritoxin diffraction data were collected at the European Synchrotron Radiation Facility (ESRF, Grenoble, France) on beamline ID29 (native data) and at the synchrotron SOLEIL (Saint-Aubin, France) on beamline PROXIMA2 (Se–Met-labeled nigritoxin) at the selenium peak with a wavelength of 0.9793 Å ($f' = -8.09$, $f'' = 6.74$). The data were processed using MOSFLM[36], Pointless[37] was used to determine the spacegroup and the data were scaled using SCALA[38, 39] within the CCP4 suite of programs (Collaborative Computational Project[40]). PrepHAData was used to convert the mtz to SHELXS format and SHELX_CDE was used to identify the 10 Se subsites through SAD phasing[41]. Finally, the program PARROT[42] provided an interpretable electron density map with an overall figure of merit of 0.83 after 15 cycles. Automatic model building with BUCCANEER[43] correctly built ~70% of the polypeptide chain. This model was then used to solve the structure of the native data set at 2.1 Å by molecular replacement and went through an iterative process of refinement using

REFMAC5[44], and model building with COOT[45] to construct the missing parts. Further data collection and refinement statistics are provided in Table 1.

To compare the folding state of the different mutant forms of nigritoxin, the polydispersity and hydrodynamic particle radius of the heterologously (and solubly) expressed proteins were measured by dynamic light scattering (Malvern Zetasizer). The wild-type protein showed no aggregation at all and the hydrodynamic radius (Rh) was measured to be 11.9 nm; the two mutant proteins showed some aggregation, but more than 98% of the volume were unaggregated particles with a Rh of 8.6 and 7.8 nm, for the "rabbit-ears" and double-histidine mutants, respectively, indicating a majority of folded protein particles in solution. In addition, the elution times in size exclusion chromatography were in the same range for the wild-type protein and the two mutants (between 51 and 73 min).

**Animal assays.** Shrimp *L. stylirostris* and *Litopenaeus vannamei* were obtained from Ifremer facilities in New Caledonia and the Roscoff Biological Station (France), respectively. Shore crabs *Carcinus maenas* and sea urchins *Paracentrotus lividus* were collected at a subtidal sampling site at the Roscoff Biological Station. Pacific oysters *Crassostrea gigas* were obtained from the Ifremer facility located at Argenton (France). Horseshoe crabs *Carcinoscorpius rotundicauda* were obtained from a commercial hatchery (Challet-Herault Aquariophilie, Nuaillé, France). The *S. littoralis* corn variant and the greater wax moth *G. mellonella* larvae (University of Montpellier) were reared on artificial diet[46] at 23 °C or with honey and pollen at 28 °C, respectively. Experiments were performed on last instar of *G. mellonella* or sixth instar (one-day-old) of *S. littoralis* larvae.

For *L. stylirostris*, *G. mellonella*, and *S. littoralis* different concentrations of nigritoxin (0.01 ng–1.2 µg protein g$^{-1}$) were tested, and each dose was injected into 10–20 animals[9, 47]. As a control, one group of 10–20 animals was injected with the protein solubilization buffer (20 mM Hepes pH 7.4, 150 mM NaCl). The surfaces of insect larvae were sterilized with 70% (vol/vol) ethanol and larvae were injected through the cuticle into the hemocoel using a Hamilton syringe with 20 µl of the appropriate dilution of nigritoxin. These experiments were performed at least twice. The calculation of LD$_{50}$ was performed using Probit Analysis[48], a type of regression used to analyze binomial response variables.

Other animals (10–20 animals per treatment) were injected using a 25-gauge needle with a solution of nigritoxin (depending on the animal 30 ng–5 µg g$^{-1}$) or protein solubilization buffer as a control. Shore crabs were injected through the arthroidial membrane at the base of the walking legs, sea urchins into the coelomic cavity near the mouth region, oysters in the adductor muscle, and horseshoe crabs through the arthrodial membrane, directly into the cardiac sinus. Experiments were performed twice.

For per-os inoculation of nigritoxin, after killing a control non-infected *L. stylirostris* shrimp, the cuticle was removed from the abdominal segment and the muscle tissues were cut into 500 mg pieces. Each portion was subsequently injected either with 5 µg nigritoxin or an equivalent volume of protein solubilization buffer and kept at 4 °C for 1 h. Acclimated shrimp were starved for 1 day before feeding individually housed shrimp with a portion of tissues either injected with nigritoxin or buffer.

For histopathological analysis, *L. stylirostris* shrimp were infected by immersion to mimic natural route of infection. Immersion challenges were performed in 100 l of filtered and aerated seawater containing $10^5$ cfu ml$^{-1}$ of the considered *vibrio* strain (virulent strain SFn1, non-virulent strain VN110[9]). Following a 2 h challenge, shrimp were transferred to 100 l tanks filled with filtered seawater, aerated and held at 27 °C. Three replicate tanks of 35 shrimp were used for each treatment. At different time points, two shrimp from each tank were randomly sampled and fixed with Davidson fixative for 48 h. Tissues were then processed following standard histological methods and stained with hematoxylin and eosin (Histalim, Montpellier, France).

For electronic transmission microscopy, *L. stylirostris* shrimp (3 replicate aquaria with 20 shrimp in each for each condition) were injected with a 100 µl volume of either a virulent strain of *V. nigripulchritudo* (SFn1, 700 cfu per animal), a non-virulent *V. nigripulchritudo* (strain VN110, 1000 cfu per animal), nigritoxin (LD$_{50}$, 3 ng g$^{-1}$ body weight) or saline as a control. At the onset of mortality, shrimp hemolymph was collected using a 25-gauge needle containing an anticoagulant solution (2% NaCl, 0.1 M glucose, 30 mM sodium citrate, 26 mM citric acid, 10 mM EDTA)[49] and cells pelleted by centrifugation (800×g for 10 min at 4 °C) before fixation.

**Electronic transmission microscopy.** Shrimp hemocytes were fixed in 5% glutaraldehyde in 0.2 M sodium cacodylate (pH 7.4) and 0.25 M sucrose[50]. The samples were then rinsed in a series of buffer solutions containing graded concentrations of sucrose and NaCl (from 0.25 M sucrose, 13 g l$^{-1}$ NaCl in 0.2 M sodium cacodylate to 0.35 M NaCl in 0.2 M sodium cacodylate) and post-fixed for 1 h at 4 °C in 1% osmium tetroxide buffered in 0.2 M of sodium cacodylate and 0.33 M NaCl. Samples were then rinsed three times for 15 min using 0.35 M NaCl and 0.2 M sodium cacodylate. Dehydration was carried out in a graded alcohol series (from 30 to 100%) and samples were finally embedded in Epon. Sections were cut using diamant knifes on a Leica ultracut UCT ultramicrotome and after staining with 2% uranyl acetate for 10 min and 2% lead citrate for 3 min, the grids were examined with a Jeol 1400 TEM (Jeol, Tokyo, Japan). Micrographs were taken using a Gatan Orius camera.

**Cell cultures**. THP-1 cells were obtained from Cell Lines Service; HeLa, Jurkat (clone E6-1), and hTERT RPE-1 cells from the American Type Culture Collection (ATCC). *Spodoptera frugiperda* Sf9 cells were obtained from ATCC and *Drosophila melanogaster* S2 cells as a gift from Dr. Lucas Waltzer (Centre de Biologie du Développement, Toulouse, France). HeLa, Jurkat, and THP-1 cells were grown at 37 °C (5% $CO_2$) in RPMI 1640 medium (Life Technologies, Gibco) with 10% fetal bovine serum (FBS). hTERT RPE-1 cells were cultured in Dulbecco's modified Eagle's medium (DMEM, Gibco) supplemented with 10% FBS. Sf9 cells were maintained in Sf-900 II SFM medium (Gibco) supplemented with 1% FBS. S2 cells were grown in 1× Schneider's Drosophila medium (Gibco) supplemented with 10% FBS and 50,000 units $l^{-1}$ of penicillin and 50,000 $\mu g\, l^{-1}$ of streptomycin. Insect cells were kept in a humidified incubator operated at 27 °C. Cell density was determined by cell counting using a Malassez counting chamber (Preciss; Strasbourg, France).

**Cell viability**. Cells were seeded at a concentration of $3 \times 10^3$ cells per well in 96 well plates with or without the indicated concentrations of proteins in triplicate. When indicated, zVAD-fmk, a general caspase inhibitor (BD Biosciences) and staurosporine, an initiator of apoptosis (S6942, 1 mM stock solution, Sigma-Aldrich) were used at a 50 and 1 μM final concentration. After incubation at 27 °C for the indicated time, 0.1 volume of Alamar Blue (Invitrogen) was added and the cells were incubated at 27 °C for an additional 3 h. Fluorescence (excitation, 530–560 nm; emission, 590 nm) was measured using a microplate reader (TECAN spark 10 M). Results are presented as a percentage of viable cells relative to experimental control (set as 100%).

**Cell cycle analysis**. Cell cycle analysis of HeLa and Jurkat cells was performed by propidium iodide (PI) staining[51]. Briefly, purified nigritoxin or an identical volume of protein solubilization buffer was added to triplicate 25 cm$^2$ flasks containing $3 \times 10^5$ HeLa or Jurkat cells in 5 ml of medium (1.2 μM final protein concentration). The cells were then incubated at 37 °C for 12 h in a humidified atmosphere of 5% carbon dioxide in air. Thereafter, HeLa cells were removed from the flasks by trypsinization. Jurkat and HeLa cells were then washed with PBS and fixed with 70% ice-cold ethanol at 4 °C for 2 h. The fixed cells were washed with PBS and incubated in the dark with PI/RNase staining buffer (BD Pharmingen) for 30 min. The percentages of a minimum of $7.0 \times 10^3$ cells at each stage of the cell cycle were quantitated based on excitation at 488 nm and emission at 670 nm (FACS Canto II Flow cytometer, Becton Dickinson, San Jose, CA, USA). The data were analyzed using FCS Express software (De Novo Software).

**Caspase activity assay**. Sf9 and S2 cell protein extracts for caspase activity were prepared from 1 ml of pelleted control or treated cells at the indicated times. Cell pellets were suspended in 200 μl ice-cold buffer A (100 mM HEPES–NaOH, pH 7.5, 10 mM dithiothreitol), homogenized by short sonication (5 s pulse, amp. 40), and centrifuged at 14,000×g for 15 min at 4 °C to obtain cytosolic extracts. Caspase assays were carried out in triplicate at 20 °C for 3 h in 1 ml buffer A, containing 100 μg protein extracts and 10 μM acetyl-DEVD-4-methyl-coumaryl-7-amide (Ac-DEVD-MCA) substrate (BD Bioscience). The fluorogenic product substrate was detected at the end of the incubation by excitation at 380 nm and emission at 480 nm with a microplate reader (TECAN spark 10 M).

**Antibodies**. The antibody against nigritoxin was developed by Eurogentec (Brussels) by injecting one rabbit with $4 \times 100$ μg purified 6His-tagged nigritoxin and collecting the serum after 28 days (1:100 for immunofluorescence and 1:1000 for Western blotting). Antibodies against the following molecules were used for immunofluorescence or Western blotting: Alexa Fluor 488 donkey α-rabbit IgG (H +L; A-21206, Invitrogen, 1:500), actin (20–33) (A5060, Sigma, 1:1000) and peroxidase-conjugated rabbit immunoglobulins (P0217, Dako, 1:5000).

**Immunofluorescence**. Cells were washed once with PBS and fixed for 30 min at RT with a solution containing 3.7% paraformaldehyde and 0.2% Triton X-100. After washing in PBS for 5 min, the cells were then incubated for 1 h at 25 °C with the anti-nigritoxin polyclonal antibody at a 1:100 dilution in PBS supplemented with 1% BSA. After incubation at 4 °C for 12 h and four times washing with PBS, cells were incubated with the anti-rabbit secondary antibody conjugated to Alexa Fluor 488 (Invitrogen) at a dilution of 1:500 in PBS containing 1% BSA for 45 min in the dark. Following three times washing with PBS, slides were mounted using VECTASHIELD Mounting Medium with DAPI (Vector Laboratories). Slides were analyzed and images collected using an inverted Leica laser-scanning confocal microscope TCS SP5 AOBS (Leica Microsystems, Heidelberg, Germany) with argon laser for illumination or using a Zeiss Axio Observer microscope (Zeiss, France). Observations were made using 63× N.A. 1.4 objective lens.

**Western blot analysis**. Reduced samples were prepared in 5× Laemmli buffer (62.5 mM Tris-HCl, pH 6.8, with 2% SDS, 20% glycerol, 5% β-mercaptoethanol, and 0.01% bromophenol blue), heated at 95 °C for 5 min, separated by SDS–PAGE and transferred onto nitrocellulose membranes using the Trans-Blot Turbo Transfer System (Bio-Rad), followed by incubation in blocking buffer (1× TBS with 0.1% Tween-20 (TBS-T), 5% milk) for 45 min. Blots were incubated for 1 h in primary antibody diluted in TBS-T (1:1000), washed three times with TBS-T, then incubated in secondary antibody diluted in TBS-T (1:5000) for 1 h. Membranes were washed three times with TBS-T, and blots were developed using the Clarity Western ECL Blotting Substrate (Bio-Rad), following manufacturer's instructions.

**Data availability**. The nigritoxin structure that supports the findings of this study has been deposited in the protein data bank (PDB) under the accession code 5M41. All other data are available from the corresponding author upon reasonable request.

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

## Acknowledgements

We are indebted to the European Synchrotron Research Facilities (ESRF, Grenoble, France) for regular access to X-ray beamlines and to all local contacts for their support during data collection at the MX beamline ID29. We also thank the French synchrotron at SOLEIL (St. Aubin) for regular access to the MX beamline PX2, and especially William Shepard, Gavin Fox, and Martin Savko (SOLEIL-St. Aubin) for valuable help during the X-ray data collection and data treatment. We thank Dr. Lucas Waltzer (Centre de Biologie du Développement, Toulouse, France) for providing the S2 cell line. We thank the staff of the station Ifremer Nouvelle Caledonie, the M3 service of the Station Biologique of Roscoff, Nathalie Desban, Florence Solari, Christophe Lambert, and Dominique Marie for technical support. We warmly thank Brigid Davis (HMS, Boston, USA) and Melanie Blokesch (EPFL, Lausanne Switzerland) for fruitful discussions and critically reading the manuscript. The present study has been supported by the ANR (11-BSV7-023-01 <<VIBRIOGEN>> and 13-ADAP-0007-01 <<OPOPOP>>).

## Author contributions

Y.L., S.C., A.J., V.B., D.A., S.L.P., and S.P. performed experiments. M.C. performed the protein structure determination and analyses. Y.L., A.G., M.C., and F.L.R. designed experiments, interpreted results, and wrote the paper.

## Additional information

**Competing interests:** The authors declare no competing financial interests.

