## [Peer Review File · Nature Communications]

Reviewers' comments:

Reviewer #1 (Remarks to the Author):

The manuscript NCOMMS-17-01502-T reports the structure and partial function of a bacterial toxin with apparent specific activity against crustaceans and hexapods (e.g. Tetraconata taxon). The crystal structure reveals the nigratoxin protein contains three domains and includes a previously undefined protein fold (e.g. "alpha/beta fold"). Whole animal injection assays and cell culture assays suggest that nigratoxin may specifically target some members of Tetraconata via an intracellular apoptotic pathway & authors suggest that nigratoxin could be important for future insecticidal control.

While the paper does provide an initial evaluation of the range of activity as well as the first structural characterization of nigratoxin, there are several critical issues and many minor problems that (in my opinion) should be addressed.

Major comments:

1. Animal assays. Authors indicate that 10-20 animals were used per treatment to test for activity of nigratoxin; however, this appears to represent only individuals (or technical replicates) within a single biological replicate. Also, Fig. 1 shows minimal/incomplete dose response data for animals injected. The current presentation makes it difficult to directly compare toxicity and specificity. Recommend presenting table showing dose response (e.g. LC50, LC95, or similar).

2. For the cell assays (both insect and human cell lines), cell mortality was used as the phenotype to gauge effectiveness of nigratoxin. Were any other "nonlethal" effects observed in the cells? That is, does nigratoxin show any other adverse effects on cells other than cell death? Also, it is important to note that just because cell lines (whether they're human or insect) may demonstrate a specific phenotype (apoptosis) in the presence of nigratoxin, doesn't necessarily mean that cells/tissues *in vivo* also behave similarly (immortalized cell lines can be great models, but they are just that). Was similar apoptotic cell death observed from cells/tissues obtained from nigratoxin-injected animals and compared with controls?

3. Lines 107-120. Assignment of function to N- and C-terminal domains of nigratoxin are problematic. Lines 107-114 describe that the function of the nigratoxin C-terminal domain is involved in cellular intoxication and specifically, line 112 states that toxicity is partially mediated by the C-terminal domain. However, Fig. 4a shows that recombinant protein lacking the N-terminal domain (as well as toxin lacking both the N-terminal & central domain) also lose cell toxicity. This could then be construed to say that toxicity is also in part mediated by the N-terminal and central domains. Based on Fig. 4a, it appears that any change had an effect and eliminated toxicity. Furthermore, line 111 claims that mutations in C-terminal domain (e.g. "delta rabbit ear" and "H, H->A, A") do not alter internalization in Sf9 cells. However, cell images in Fig. S7 for these mutant proteins appear different than those of wild-type protein (particularly at 12 h). Hence, the results shown in Fig. S7 actually suggests that the C-terminal domain may also be important for internalization. The function of the N-terminal domain is claimed to involve internalization of the toxin. However, the key figures shown in Fig. 4b are in doubt, as the last three of the bottom panels for 12 h (Central + C-term, C-term, and Buffer) are exact replicate images. Please explain. Furthermore, lines 116-120 and immunoblot (Fig. S8) are not helpful to discern function of nigratoxin domains. Fig. S8 shows Western blot of recombinant nigratoxins produced in BL21 (e.g. *E. coli*). How does this pertain to toxin localization or function in Sf9 cells?

4. Lines 125-133. Paper does not adequately show conserved folding/structural comparisons between Afp18 and nigratoxin. Namely, Fig. S10 only shows rudimentary nigratoxin structural

features and does not show Afp18 structure for comparison (overlay showing conserved structural features should be included). Conclusion that nigrifitin and Afp18 have distinct mechanisms (lines 132-133) is not adequately supported in this paper.

5. Justification for future insecticidal applications is relatively weak. Namely, only two insect species tested here as well as many conclusions that are based on 1-2 lines of immortalized insect cell cultures. Furthermore, delivery of intracellular toxin will be problematic (as indicated by authors). Hence, although one must start somewhere, the paper appears to be "reaching" for application with the supplied data.

Minor comments:

1. Line 19. Recommend using "due to the evolution of resistance" rather than "buildup of resistance".
2. Line 36. Should be "humans" rather "human"
3. Line 37. Insert commas after "fever" & "crops". Also, insect pests are also responsible for damage to human structures.
4. Line 39. Insert "and other non-target organisms" as humans were not only animals impacted by overuse of DDT.
5. Lines 39-40. It can be argued that the off-target effects of DDT not only led to efforts to development of more specific and environmentally benign insecticides, but also the overall recognition that there is a need to limit potential unwanted exposure of particularly toxic, broad-spectrum compounds in the environment.
6. Lines 40-42. This sentence seems to suggest that chemical insecticides are no longer important components of successful pest management strategies. It is true that Bt toxins (e.g. Bt crops) dominate the "biologic" insecticides, but chemical pesticides do remain essential components of many agricultural systems.
7. Lines 43-44. Revise sentence to "and commercialized Bt toxins are not toxic to some key insect pests, including the hemipterans."
8. Lines 44-45. Yes, these have been identified, but are not yet commercially available.
9. Line 46. Start new paragraph following "snowdrop lectins."
10. Lines 49-52. While this statement may be true, couldn't pathways and toxins also exist that target more distantly related invertebrates and humans? It might carry more weight if authors present evolutionary distances or clock showing that actual distance between crustaceans and insects is much smaller than with other higher animals (show Fig. 1 with actual distances rather than arbitrary ones). Alternatively, doesn't the literature suggest that a more common approach is for the evolution of highly specific toxins (e.g. Bt toxins) that target few select terrestrial insects and not other animals?
11. Line 58. Lethality of animals observed. What about other non-lethal effects?
12. Line 61-62. Suggest changing to "Nigrifitin did not cause larval mortality...".
13. Line 72. Replace "did not modify cell viability" with "did not target" or "did not kill".
14. Lines 82-83. Is "resistance" is the correct term here? I suggest revising sentence to something like "Immunofluorescence analyses show that while two types of cultured insect cells are susceptible to nigrifitin, human cells may be refractory to the toxin due to the inability of the toxin to bind and/or internalize (Fig. 2c)."
15. Line 85. Insert "Sf9" after "intoxicated" and insert "protein" after "apparent".
16. Line 94. Replace "stick" with "project".
17. Line 100. Replace "potentially is involved in biological function" with "has yet to have a defined biological function".
18. Lines 102-103. Authors claim that nigrifitin structure contains overall new fold (e.g. alpha/beta class). However, neither Fig. 3, Fig. S10, nor the text adequately describes the novelty or significance of this new fold. Why is it named alpha/beta?
19. Line 117. "Western" should be capitalized.
20. Line 134. Uncertain what is meant by "purified"?
21. Line 150. "clonings" should be "cloning".

22. Line 153. Revise sentence to "Histidine (6His)-tagged fusion proteins were purified by...".
23. Line 155. What is source of nigratoxin protein for antibody production?
24. Fig. 1. Animals in black should show 0% mortality after 300 ng per g dose.
25. Fig. 1 (line 294). Revise sentence to "# indicates the natural target of nigratoxin."
26. Fig. 3 (line 324). Replace "sticking out" with "projecting".
27. Fig. 3c. Define specific residues of interest within figure by using color. That is, highlight and number residues in structure that are shown highlighted in red below. It is not clear if these His and Arg residues lie in close proximity in Fig. 3c. Lines 328-329 are written as pure speculation, as the "large patch" itself was not tested for any function.
28. Fig. 4a. Watch overlay of graphics for symbols, as it appears that some are masking others.
29. Supplemental Information.
 - a. Line 19. Do you mean CDS of gene (e.g. cDNA)?
 - b. Line 23. Revise for clarity.
 - c. Line 33. See comment 29a above.
 - d. Lines 37-39. Incomplete sentence.
 - e. Line 40. Replace "DNase" with "DNase".
 - f. Lines 47-48. Watch spacing between 150 mM and pH 7.4.
 - g. Lines 48-49. If immunoblot is retained (Fig. S8), why not include stained SDS-PAGE gel as well?
 - h. Lines 91, 92, 165. Watch use of "," instead of ".".
 - i. Lines 102-103. Indicate biological replication (see major comment 1 above).
 - j. Line 110. Insert "C" following "23 degrees".
 - k. Lines 110-113. Where were injections performed on insect larvae?
 - l. Line 134. Replace "c" with "C".
 - m. Line 139 (Fig. 2b & S4). Describe biological replication performed for all caspase assays.
 - n. Line 157. Describe approximate number of hours.
30. Suppl. Fig. S1. Line 202. Provide reference for the Alamar Blue assay.
31. Suppl. Fig. S1. Explain why S2 cell viability is only ~80% in Buffer. Seems to suggest that cells may not be healthy.
32. Suppl. Fig. S2. Not sure figure shows all steps of apoptotic cell death as described in text?

Criteria for publication:

Technically sound: Partly (further clarification is needed to confirm adequate biological replication was conducted and lacks adequate functional validation of the N- & C-termini of nigratoxin – see comments above).

Provides strong evidence for conclusions: Partly (limited use of representative animals and cell lines could be strengthened; again, function of domains not clearly shown; and relative narrow spectrum of activity and lack of non-lethal or non-specific interactions provides relatively weak relevance for practical use as insecticide).

Results are novel: Yes (although, identification of nigratoxin was previously shown)

Important to field: Perhaps (it remains to be seen how this study will impact the understanding of toxin activity/specificity and/or if nigratoxin will have any practical application, namely as a novel pesticide/insecticide).

Reviewer #2 (Remarks to the Author):

This manuscript describes further structural characterization of a novel toxin, nigratoxin, produced by a bacterial pathogen of shrimps. Interestingly, nigratoxin is shown to have insecticidal activity making its characterization of general interest due to its potential use for controlling insect pests.

Furthermore, nigratoxin showed toxicity to insect cell lines as Sf9 or S2 but not to mammalian cell lines confirming its specificity. By using insect cell lines, the authors show that nigratoxin induces cell death by cell internalization triggering cell death by apoptosis. Solving the crystal structure of nigratoxin revealed a three-domain structure composed of a N-terminal domain, a central domain and a c-terminal domain. N-terminal domain and part of the central part domain show structural similarity with another partially characterized toxin, Afp18. The c-terminal domain of nigratoxin shows a novel three-dimensional fold where two extended loops and a histidine rich pocket were shown to be important for toxicity to Sf9 cells by the characterization of the c-terminal domain mutations and truncated variants of the toxin. The data shows that the n-terminal domain is involved in cell specificity and cell internalization while the c-terminal domain is responsible for triggering cell death. This is an interesting report that highlights the characterization of novel insecticidal proteins with potential for pest control. The authors should discuss on potential cellular targets of nigratoxin. Also, the authors should tune down the statement of the potential use of nigratoxin for pest control since the insecticidal activity of nigratoxin is lower than that shown against *Litopenaeus stylirostris* and toxicity against insects was not observed by feeding. The authors state that this is a potent toxin but it is difficult to judge the potential use of this protein for pest control without defining the potency (LD50) against different insect pests as *G. mellonella* and *S. littoralis*, and compare with published reports for other insecticidal toxins.

1. Page 3 line 46. It is stated that Bt toxins do not control hemipteran pests as a drawback for the use of Bt toxins. However, it is not known if nigratoxin is active against hemipteran pests. This could be interesting to determine. It could be discuss that the toxicity of nigratoxin against hemipteran and against Bt resistant lepidopteran and coleopteran insects would reveal if nigratoxin could be an alternative for pest control.

2. Line 61. References 24 and 25 of manuscript are cited as venom toxins with insecticidal potential but those manuscripts did not perform toxicity against insects.

3. Line 65-66. The authors claim that the results demonstrate that nigratoxin is a potent toxin that is highly specific. As mentioned above LD50 to the lepidopteran insects would be informative.

Reviewer #3 (Remarks to the Author):

The manuscript by Labreuche and colleagues reports a novel crystal structure, that of nigratoxin. While the structure does not seem to inform the understanding of nigratoxin mechanism, they do present data to show that nigratoxin is toxic across the tetraconata, a family that includes crustaceans and insects. The paper is well written, the crystallographic analysis appears solid, and for the most part the conclusions are well supported by the data.

Principle concerns:

Since this is a new and novel structure, I think the authors should do more to describe it. For example, Fig. 3 is too small. Why not point out the residues that coordinate the Mg²⁺. Mutate those? Fig. 3c inset is not very effective. Show the side chains- especially point out the two that are mutated. Show electrostatic surface of whole thing. Show some representative electron density in supplement.

Need to present evidence that the mutants are still folded. Western blot is insufficient as this is in SDS. This is especially relevant for the delta rabbit ear mutant. Further, there are some single point mutants for the Trp and the Arg of the 'rabbit ear' groove, described in the methods but no data for these. These sound interesting.

Minor concerns:

S5a/b: In order to distinguish between what accumulates in the cytosol and what could remain bound to the outside, one should test lysates from 0 minutes intoxication to show all signal

washed off.

Fig. 4 legend: numbering of histidines does not match Fig. 3c.

Reviewer #1

Major comments:

Animal assays. Authors indicate that 10-20 animals were used per treatment to test for activity of nigrifoxin; however, this appears to represent only individuals (or technical replicates) within a single biological replicate. Also, Fig. 1 shows minimal/incomplete dose response data for animals injected. The current presentation makes it difficult to directly compare toxicity and specificity. Recommend presenting table showing dose response (e.g. LC50, LC95, or similar).

We modified the Figure 1 to show a) the dose injected to different species representative of diverse phyla within the Bilateria. b) the evolutionary distances in substitutions/site between representative species within the Bilateria. The LD₅₀ was precisely determined for *L. stylirostris* shrimp and *G. mellonella* using Probit analysis and the results are now shown in Figure S1. Experimental infections were performed at least twice as mentioned in the Extended data, lines 116 and 125.

For the cell assays (both insect and human cell lines), cell mortality was used as the phenotype to gauge effectiveness of nigrifoxin. Were any other “nonlethal” effects observed in the cells? That is, does nigrifoxin show any other adverse effects on cells other than cell death? Also, it is important to note that just because cell lines (whether they’re human or insect) may demonstrate a specific phenotype (apoptosis) in the presence of nigrifoxin, doesn’t necessarily mean that cells/tissues in vivo also behave similarly (immortalized cell lines can be great models, but they are just that). Was similar apoptotic cell death observed from cells/tissues obtained from nigrifoxin-injected animals and compared with controls?

We performed a cell cycle analysis by flow cytometry of HeLa and Jurkat cells following incubation with nigrifoxin (used at the highest tested dose *i.e.* 1.2 μM) for 12 h. We did not observe any adverse effect of nigrifoxin on cell cycle progression for these two cell types compared to control cells. These data are now included in the revised version of the manuscript (Extended data Table 4). We next performed a histopathological analysis of shrimp transiently immersed into *V. nigripulchritudo*-contaminated waters, as we wanted to model the natural route of infection. Pathological changes observed in infected shrimp were solely characterized by abnormal hemocytes, many of them being karyopyknotic, dispersed throughout the gills, heart and digestive tubules. Such cellular alterations were not seen in control animals. Based on these data, we next performed a transmission electron microscopy study focused on the cytopathogenic effects induced in shrimp hemocytes *in vivo* by injection of either *V. nigripulchritudo* or nigrifoxin. We observed that nigrifoxin induces the same alterations as these occurring during infection by *V. nigripulchritudo*, *i.e.* hemocytes showing membrane disruption, chromatin condensation (pyknosis) and nucleus fragmentation (karyorrhexis). Altogether these experiments show that experimentally challenged shrimp using different infection models or insect cells exposed to the toxin *in vitro* present similar morphological signs of apoptosis. Corresponding data (Supplementary Figure 2 and 3) are now included in the revised version.

Lines 107-120. Assignment of function to N- and C-terminal domains of nigrifoxin are problematic. Lines 107-114 describe that the function of the nigrifoxin C-terminal domain is involved in cellular intoxication and specifically, line 112 states that toxicity is partially mediated by the C-terminal domain. However, Fig. 4a shows that recombinant protein lacking the N-terminal domain (as well as toxin lacking both the N-terminal & central

domain) also lose cell toxicity. This could then be construed to say that toxicity is also in part mediated by the N-terminal and central domains.

The protein lacking the N-terminal domain is not internalized as can be seen in the Figure 5b. It is therefore impossible to unambiguously state whether the N-terminal domain contributes to the toxicity effect or not. Consequently, we wrote in line 161 that the toxicity is mediated at least in part through nigrیتoxin's C terminal domain. We also replaced the last sentence at the end of the paragraph (line172) “while the C-terminus triggers cell death through apoptosis” by “C-terminal domain is necessary to induce cell death through apoptosis.”

Based on Fig. 4a, it appears that any change had an effect and eliminated toxicity.

We added in the Figure 5a the result obtained with a double substitution with alanine of the two amino acids (Arg512 and Trp540) suspected to be involved in the stabilization of the protruding loops that did not affect toxicity to Sf9 cells (line 156).

Furthermore, line 111 claims that mutations in C-terminal domain (e.g. “delta rabbit ear” and “H, H->A, A”) do not alter internalization in Sf9 cells. However, cell images in Fig. S7 for these mutant proteins appear different than those of wild-type protein (particularly at 12 h).

We performed this experiment several time and now compile all immunofluorescence pictures in one figure (now Fig.5b). Note that the signal at 12h of R,W mutant and wild type are difficult to compare with other versions of the protein because they cause dramatic cell and nuclei damages (white arrow on Fig5b). We agree that H,H mutant appears slightly different at 12h and thus change the text (line 159) by “None of these mutations prevented protein internalization” as can be clearly appreciated at the 1h time-point.

Hence, the results shown in Fig. S7 actually suggests that the C-terminal domain may also be important for internalization.

We truly believe that the data on Figure 5b showing that the N-term or central + N-term mutants give a similar signal of protein in the cytosol than the wild type, are supportive of the notion that the C-terminal domain is not essential for internalization.

The function of the N-terminal domain is claimed to involve internalization of the toxin. However, the key figures shown in Fig. 4b are in doubt, as the last three of the bottom panels for 12 h (Central + C-term, C-term, and Buffer) are exact replicate images. Please explain.

We apologized for this mistake, images were added as placeholder initially and unfortunately they were not replaced by the final data. Experiments have been performed several times and we hope that images set in Fig.5b fulfill the reviewer attempts.

Furthermore, lines 116-120 and immunoblot (Fig. S8) are not helpful to discern function of nigrیتoxin domains. Fig. S8 shows Western blot of recombinant nigrیتoxins produced in BL21 (e.g. E. coli). How does this pertain to toxin localization or function in Sf9 cells?

This western blot and (and now SDS PAGE as further requested), are presented to show that all truncated forms are stable and are detected by the antibodies. We changed the sentence (line 166) by « The truncated forms of nigrیتoxin were similarly detected by SDS PAGE and Western blot with polyclonal anti-nigrیتoxin antibodies, showing that these results were not due to protein instability or lack of detection by the antibodies (Fig.S12).”

4. Lines 125-133. Paper does not adequately show conserved folding/structural comparisons between Afp18 and nigratoxin. Namely, Fig. S10 only shows rudimentary nigratoxin structural features and does not show Afp18 structure for comparison (overlay showing conserved structural features should be included). Conclusion that nigratoxin and Afp18 have distinct mechanisms (lines 132-133) is not adequately supported in this paper.

We regret for perhaps not having been clear enough on this point. We would like to ask the reviewer to consider that while Fig S10 (now Fig.S14) shows the classical ribbon-representation, the more detailed topology of nigratoxin are indeed depicted in Figure S11. However, it is important to note here that there is no 3D structure determination of Afp18 or even of the domain of Afp18 that is homologous to a part of nigratoxin. Nevertheless, in structural biology, it is granted that with more than 40% sequence identity the fold of those domains is structurally (and evolutionarily) related, as highlighted by the multiple sequence alignment in Figure S15. So the only ‘overlay’ that can be given by now is with the sequence alignment in that figure, in which the secondary structure elements of nigratoxin are given above the sequences.

5. Justification for future insecticidal applications is relatively weak. Namely, only two insect species tested here as well as many conclusions that are based on 1-2 lines of immortalized insect cell cultures. Furthermore, delivery of intracellular toxin will be problematic (as indicated by authors). Hence, although one must start somewhere, the paper appears to be “reaching” for application with the supplied data.

We completely agree with these comments and changed part of the abstract, introduction, and discussion of our manuscript to focus more on fundamental question regarding the nigratoxin tropism for Tetraconata and mechanism of action.

Minor comments:

Line 19. Recommend using “due to the evolution of resistance” rather than “buildup of resistance”.

This sentence is not in the text anymore

2. Line 36. Should be “humans” rather “human”

This sentence is not in the text anymore

3. Line 37. Insert commas after “fever” & “crops”. Also, insect pests are also responsible for damage to human structures.

This sentence is not in the text anymore

4. Line 39. Insert “and other non-target organisms” as humans were not only animals impacted by overuse of DDT.

This sentence is not in the text anymore

5. Lines 39-40. It can be argued that the off-target effects of DDT not only led to efforts to development of more specific and environmentally benign insecticides, but also the overall recognition that there is a need to limit potential unwanted exposure of particularly toxic, broad-spectrum compounds in the environment.

This sentence is not in the text anymore

6. Lines 40-42. This sentence seems to suggest that chemical insecticides are no longer important components of successful pest management strategies. It is true that Bt toxins (e.g. Bt crops) dominate the “biologic” insecticides, but chemical pesticides do remain essential components of many agricultural systems.

This sentence is not in the text anymore

7. Lines 43-44. Revise sentence to “and commercialized Bt toxins are not toxic to some key insect pests, including the hemipterans.”

This sentence has been moved to the discussion and changed according to the reviewer demand (line 236)

8. Lines 44-45. Yes, these have been identified, but are not yet commercially available.

This sentence is not in the text anymore

9. Line 46. Start new paragraph following “snowdrop lectins.”

This sentence is not in the text anymore

10. Lines 49-52. While this statement may be true, couldn’t pathways and toxins also exist that target more distantly related invertebrates and humans? It might carry more weight if authors present evolutionary distances or clock showing that actual distance between crustaceans and insects is much smaller than with other higher animals (show Fig. 1 with actual distances rather than arbitrary ones). Alternatively, doesn’t the literature suggest that a more common approach is for the evolution of highly specific toxins (e.g. Bt toxins) that target few select terrestrial insects and not other animals?

We acknowledge the reviewer for this very helpful remark and changed the Figure 1 to show evolutionary distance between tested animals. In addition in the last paragraph of the discussion (line 235-238) we wrote “Targeting a broad range of insect species represents an alternative to the narrow activity spectrum of commercialized biological insecticides such as *Bacillus thuringiensis* toxins (Bt)²⁸ which are not toxic to some key insect pests, including the hemipterans (e.g. sap-sucking insects) pests³². We speculate that nigratoxin holds promise as a novel insecticidal protein but delivery systems have to be devised to access its target site *i.e.* the animal circulatory system.”

11. Line 58. Lethality of animals observed. What about other non-lethal effects?

In addition of the absence of lethal effect, control injections had no observable changes in, for example, the phenotype or appearance and behavior of tested animals. This is now indicated in the revised version (line 93).

12. Line 61-62. Suggest changing to “Nigratoxin did not cause larval mortality...”.

Done line 99

13. Line 72. Replace “did not modify cell viability” with “did not target” or “did not kill”.

Done line 110

14. Lines 82-83. Is “resistance” is the correct term here? I suggest revising sentence to something like “Immunofluorescence analyses show that while two types of cultured insect cells are susceptible to nigratoxin, human cells may be refractory to the toxin due to the inability of the toxin to bind and/or internalize (Fig. 2c).”

Modifications were done as suggested by the referee (line 120-122)

15. Line 85. Insert “Sf9 after “intoxicated” and insert “protein” after “apparent”.

Done line 125

16. Line 94. Replace “stick” with “project”.

Done line 138

17. Line 100. Replace “potentially is involved in biological function” with “has yet to have a defined biological function”.

Done line 143

18. Lines 102-103. Authors claim that nigratoxin structure contains overall new fold (e.g. alpha/beta class). However, neither Fig. 3, Fig. S10, nor the text adequately describes the novelty or significance of this new fold. Why is it named alpha/beta?

The novelty of the fold, as described in the text resides in the three domain-organization and that, even with each module individually, no similarity is detected to other structural folds. The attribution alpha/beta class (as defined by CATH fold classification, reference now added in the manuscript, line 149) by definition is because the fold contains both alpha-helices and beta-strands and sheets. We have now added a comment to the main text indicating the significance of the new fold as being, for the moment, an “orphan fold” (line 147). Consequently, no potential active site can be identified by dedicated programs or structural comparison. In addition, novel structural features are definitely described in lines 136-144.

19. Line 117. “Western” should be capitalized.

Done line 167

20. Line 134. Uncertain what is meant by “purified”?

We removed “as purified protein” from the sentence to make it clearer

21. Line 150. “clonings” should be “cloning”.

Done line 243

22. Line 153. Revise sentence to “Histidine (6His)-tagged fusion proteins were purified by...”.

Done line 246

23. Line 155. What is source of nigratoxin protein for antibody production?

The 6His-tagged nigratoxin purified by affinity and size exclusion chromatography was used in a rabbit immunization protocol for polyclonal antibody production, as described in the extended data (line 202-207)

24. Fig. 1. Animals in black should show 0% mortality after 300 ng per g dose.

We hope that the Figure 1 improvement will fulfill the reviewer questions.

25. Fig. 1 (line 294). Revise sentence to “# indicates the natural target of nigratoxin.”

Not relevant in the new Figure 1

26. Fig. 3 (line 324). Replace “sticking out” with “projecting”.

Done

27. Fig. 3c. Define specific residues of interest within figure by using color. That is, highlight and number residues in structure that are shown highlighted in red below. It is not clear if these His and Arg residues lie in close proximity in Fig. 3c.

We have modified Figure 3c according to the comments of the reviewer – the different residues in the ‘patch’ are given the same color-code as in the sequence.

Lines 328-329 are written as pure speculation, as the “large patch” itself was not tested for any function.

We are sorry but we do not agree with the referee. Structural analyses showed that 4 histidine residues and one arginine are concentrated in a groove forming a large patch at the protein surface. When we performed replacement with alanine of two of these 4 histidine residues (His598 and His650) to destabilize this patch, the toxicity of the resulting mutated protein towards Sf9 cells was completely abolished, showing that this region contributes to the biological activity of the nigratoxin.

28. Fig. 4a. Watch overlay of graphics for symbols, as it appears that some are masking others.

This figure has been modified (now Fig.5)

29. Supplemental Information.

a. Line 19. Do you mean CDS of gene (e.g. cDNA)?

We mean CDS and add it to text line 19

b. Line 23. Revise for clarity.

We modified the sentence as follows : “The amplicon was digested by *Bam*HI and *Xho*I and cloned into the pFO4 plasmid (modified from pET15b, Novagen, USA) that contains a His6-tag at the N-terminal for affinity purification”

c. Line 33. See comment 29a above.

See our above response

d. Lines 37-39. Incomplete sentence.

The sentence was completed by: “was added and incubated stirring at 4°C for 10 min.”

e. Line 40. Replace “DNase” with “DNase”.

Done

f. Lines 47-48. Watch spacing between 150 mM and pH 7.4.

Done

g. Lines 48-49. If immunoblot is retained (Fig. S8), why not include stained SDS-PAGE gel as well?

The SDS PAGE is now include in FigS8 (now S12)

h. Lines 91, 92, 165. Watch use of “,” instead of “.”.

Done

Lines 102-103. Indicate biological replication (see major comment 1 above).

Experimental infections were performed at least twice as now mentioned in the Extended data, lines 116 and 125.

j. Line 110. Insert “C” following “23 degrees”.

Done line 107

k. Lines 110-113. Where were injections performed on insect larvae?

“larvae were injected through the cuticle into the haemocoel using a Hamilton syringe with 20 µl of the appropriate dilution of nigratoxin », line 115

l. Line 134. Replace “c” with “C”.

Done line 174

m. Line 139 (Fig. 2b & S4). Describe biological replication performed for all caspase assays.

Done, “Caspase assays were carried out in triplicate at 20 °C” line 197

n. Line 157. Describe approximate number of hours.

Done, “After incubation at 4°C for 12 h”, line 213

30. Suppl. Fig. S1. Line 202. Provide reference for the Alamar Blue assay.

Done, The cytotoxicity was monitored using Alamar Blue (Invitrogen) assay. Line 288 (now supplementary Figure 4)

31. Suppl. Fig. S1. Explain why S2 cell viability is only ~80% in Buffer. Seems to suggest that cells may not be healthy.

Now supplementary Figure 4: we agree that the buffer has a slight but not significant and reproductive effect on S2, showing the importance of comparing the mock with the buffer in addition to the tested conditions.

32. Suppl. Fig. S2. Not sure figure shows all steps of apoptotic cell death as described in text?

In the main text the sentence has been changed by “In insect cells, nigratoxin treatment was associated with cell shrinkage, blebbing, vacuolization and DNA fragmentation and condensation (supplementary Figure 4) suggesting (line 113) the occurrence of apoptotic cell death”

Reviewer #2

The authors should discuss on potential cellular targets of nigratoxin.

We now discuss the potential cellular target of nigratoxin in a paragraph within the discussion that has been deeply modified.

Also, the authors should tune down the statement of the potential use of nigratoxin for pest control since the insecticidal activity of nigratoxin is lower than that shown against *Litopenaeus stylirostris* and toxicity against insects was not observed by feeding.

We agree with the reviewer and changed part of the abstract, introduction and the discussion to tune down the speculation regarding nigratoxin as biological insecticide.

The authors state that this is a potent toxin but it is difficult to judge the potential use of this protein for pest control without defining the potency (LD50) against different insect pests as *G. mellonella* and *S.littoralis*, and compare with published reports for other insecticidal toxins.

We defined the LD50 for *L. stylirostris* (40 fmol.g⁻¹) and *G. mellonella* (3 pmol.g⁻¹) using Probit analysis. The dose appears to be of the same order of magnitude as that of other toxins tested in marine invertebrates and in insects. The dose required to kill *Spodoptera littoralis* appears higher (9 pmol.g⁻¹) but this is also the case for other toxins tested by injection in this animal (Givaudan and coll, pers.com). Indeed, *S. littoralis* is known to resist better to toxin injection.

Page 3 line 46. It is stated that Bt toxins do not control hemipteran pests as a drawback for the use of Bt toxins. However, it is not known if nigratoxin is active against hemipteran pests. This could be interesting to determine. It could be discuss that the toxicity of nigratoxin against hemipteran and against Bt resistant lepidopteran and coleopteran insects would reveal if nigratoxin could be an alternative for pest control.

We agree with the reviewer that more animal species, in particular insects, must be tested in the future (line 197). According to the reviewer requests, our manuscript has been rewritten to focus on nigratoxin specificity and mechanisms and prevent too much speculation on application.

Line 61. References 24 and 25 of manuscript are cited as venom toxins with insecticidal potential but those manuscripts did not perform toxicity against insects.

We agree with the reviewer and re-wrote the sentence as follows: “All crustaceans died within 24 h following injection of 30 ng.g⁻¹ (400 fmol.g⁻¹) body weight of protein (Fig.1a) a dose comparable to that of venom toxins with activity on marine crustaceans^{14,15}”(line 96).

3. Line 65-66. The authors claim that the results demonstrate that nigratoxin is a potent toxin that is highly specific. As mentioned above LD50 to the lepidopteran insects would be informative.

The LD50 has been calculated using Probit analyses for *G. mellonella* and is now presented in Figure 1 and supplementary Figure 1.

Reviewer #3

Principle concerns:

Since this is a new and novel structure, I think the authors should do more to describe it. For example, Fig. 3 is too small. Why not point out the residues that coordinate the Mg²⁺. Mutate those? Fig. 3c inset is not very effective. Show the side chains- especially point out the two that are mutated. Show electrostatic surface of whole thing. Show some representative electron density in supplement.

In view of space limits, it is difficult to make larger figures – the figure is supposed to give an overall view. We have instead added some figures showing details, such the Mg²⁺ coordination (Figure 4) and the electron density of the “rabbit ears” (Supplementary Figure 10). Since structural analyses revealed a surface loop in the N-ter domain containing a Mg²⁺, we were indeed interested in the potential role of this ion in the nigrigrotoxin biological activity. Before deciding to perform site-directed mutagenesis of the residues coordinating this ion, we first assessed its role by incubating Sf9 cells with nigrigrotoxin in the presence or absence of EDTA, a chelating agent widely used to sequester di- and trivalent metal ions such as Mg²⁺ and subsequently determining cell viability. EDTA was added at two different final concentrations (100 μM and 1 mM), determined as innocuous to Sf9 cell viability. However, we did not observe any significant change in nigrigrotoxin toxic effects following addition of this chelating agent for both tested concentrations. This indicates that the Mg²⁺ ion has before all a structural role. We also injected shrimp with *V.nigripulchritudo* supernatant (containing the nigrigrotoxin) in the presence or absence of EDTA (50 mM, final concentration). In this case too, addition of EDTA did not reduce the toxicity of the bacterial supernatant, excluding a role of this ion in the toxicity. These considerations are now mentioned in the revised version of the manuscript (line 133-136).

Need to present evidence that the mutants are still folded. Western blot is insufficient as this is in SDS. This is especially relevant for the delta rabbit ear mutant.

To compare the folding state of the different mutant forms of nigrigrotoxin, the polydispersity and hydrodynamic particle radius of the heterologously (and solubly) expressed proteins were measured by dynamic light scattering (Malvern Zetasizer). The wild type protein showed no aggregation at all and the hydrodynamic radius (R_h) was measured to be 11.9 nm; the two mutant proteins showed some aggregation, but more than 98% of the volume were unaggregated particles with a R_h of 8.6 and 7.8 nm, for the « rabbit-ear » and double-histidine mutants, respectively, indicating a majority of folded protein particles in solution. In addition the elution times in size exclusion chromatography were in the same range for the wild type protein and the two mutants (between 51 and 73 min). This result has been added in the extended data (line 90-98) and we added in the main text (Line 160) “None of these mutations prevented protein internalization (Fig.5b) and altered the folding of the proteins as measured by dynamic scattering (Extended data).”

Further, there are some single point mutants for the Trp and the Arg of the ‘rabbit ear’ groove, described in the methods but no data for these. These sound interesting.

We created alanine substitutions by site-directed mutagenesis of the two residues (Arg512 -> Ala and Trp540->Ala) suspected to be involved in the stabilization of the protruding loops observed in the C-terminal domain of nigrigrotoxin and tested the toxicity of this mutant on insect cell viability. Alanine was chosen because it is considered to be the most neutral amino acid and a side chain of alanine is the least voluminous of all the amino acids. Furthermore, it is not highly hydrophobic and has no charge. However this double mutant revealed as toxic as the wild-type nigrigrotoxin, whereas the total elimination of these loops abolished the toxicity,

demonstrating their role in the biological activity and suggesting that alanine substitutions of the Trp and the Arg were insufficient to destabilize this structural feature. We agree with the referee and added these data in the revised version of the manuscript (Figure 5, line 155).

Minor concerns:

S5a/b: In order to distinguish between what accumulates in the cytosol and what could remain bound to the outside, one should test lysates from 0 minutes intoxication to show all signal washed off.

We agree with the reviewer that 0h incubation would be an additional interesting control, but the aim of this figure (now supplementary Figure 8) was to prove that the protein is not processed/cleaved in the cytosol with time (as are some bacterial toxins).

Fig. 4 legend: numbering of histidines does not match Fig. 3c.

We apologize for this mistake that is now corrected in the revised version of the manuscript.

REVIEWERS' COMMENTS:

Reviewer #1 (Remarks to the Author):

The revised manuscript NCOMMS-17-01502A adequately addresses critical issues raised in my previous review. I do have few additional comments:

Major comment:

1. The inclusion of new data/figures showing histopathology of virulent bacteria and nigratoxin within shrimp hemocytes is very much appreciated. However, it seems that paper would be improved (namely the questions that are brought forward in discussion, lines 223-232) if similar tests were included against insect (G. mellonella and S. littoralis, or other model insect) hemocytes.
2. Supplemental methods, lines 51-54. It is unclear if recombinant nigratoxin used for structural determination (and/or functional assays) retained the histidine tag used for affinity purification. If so, does the tag impact overall structure (and/or activity of protein)?

Minor comments:

1. Line 74. Replace "injected to..." with "injected into.."
2. Line 174. Use of "significant" indicates statistical difference was observed. Suggest using "substantial" or other non-statistical term here.
3. Line 199. Unsure what is meant by "a crustacean-specific trait". Perhaps sentence should state "confirmed that a crustacean-active protein has similar activity against insect cells..."
4. Line 214-215. Suggest revising sentence to "The primary target of nigratoxin may within the cytoplasm and could involve caspases..."
5. Line 443. Replace "3" with "three"
6. Line 327. This figure is incorrectly labeled as Figure S7 (should be Supplemental Figure 8).

Reviewer #2 (Remarks to the Author):

The authors have address and modify the manuscript according to the reviewers suggestions. I only suggest that the LD50 data against Spodoptera litura is mention in the text as the one against Galleria mellonella.

Reviewer #3 (Remarks to the Author):

The authors have provided new figures and information which address my previous set of comments.

They have also been responsive to the comments made by the other reviewers.

Reviewer #1

Major comment:

1. The inclusion of new data/figures showing histopathology of virulent bacteria and nigratoxin within shrimp hemocytes is very much appreciated. However, it seems that paper would be improved (namely the questions that are brought forward in discussion, lines 223-232) if similar tests were included against insect (*G. mellonella* and *S. littoralis*, or other model insect) hemocytes.

We are pleased that the reviewer#1 appreciates the demonstration that Nigratoxin mimics the V. nigripulchritudo effect on shrimp hemocytes. However similar analyses using insect models are not possible, as Vibrio (a halophile bacteria) is not able to infect terrestrial insect or be incubated with insect cell lines. We agree (and discuss) in the discussion section that the major perspective of the present work is to understand the fine molecular mechanisms of Nigratoxin action, i.e. identification of a receptor and intracellular route of the toxin in the eukaryotic target cell. This will be explored in the future using insect cell lines as shrimp cell line are not available.

2. Supplemental methods, lines 51-54. It is unclear if recombinant nigratoxin used for structural determination (and/or functional assays) retained the histidine tag used for affinity purification. If so, does the tag impact overall structure (and/or activity of protein)?

The His-tag is retained in the recombinant protein but does not interfere with the structure of the protein since it is disordered (or floppy), as other residues of the N-terminal. We can't see the structure from amino acid 18 (the signal peptide has been removed in the recombinant protein). In addition all in vivo and in vitro tests have been performed with recombinant-His-tag containing protein and hence demonstrated to be active. We added in the supplemental methods, end of the paragraph Purification of recombinant protein the following sentence : "The recombinant proteins used for structural determination and functional assays retained the histidine tag without impacting the structure and function of the protein."

Minor comments:

1. Line 74. Replace "injected to..." with "injected into..."

Done

2. Line 174. Use of "significant" indicates statistical difference was observed. Suggest using "substantial" or other non-statistical term here.

Done

3. Line 199. Unsure what is meant by "a crustacean-specific trait". Perhaps sentence should state "confirmed that a crustacean-active protein has similar activity against insect cells..."

We propose to change the sentence by "Although it is necessary to confirm these data by testing more animal species in experimental infection, further exploration of the in vitro

cytotoxicity of nigratoxin confirmed that a crustacean-specific receptor and/or means of translocation, is conserved in insect cells and targeted by the N-terminal domain of nigratoxin which is required for its internalization”

4. Line 214-215. Suggest revising sentence to “The primary target of nigratoxin may within the cytoplasm and could involve caspases...”

Done

5. Line 443. Replace “3” with “three”

Done

6. Line 327. This figure is incorrectly labeled as Figure S7 (should be Supplemental Figure 8).

Done

Reviewer #2

The authors have address and modify the manuscript according to the reviewers suggestions. I only suggest that the LD50 data against *Spodoptera litura* is mention in the text as the one against *Galleria mellonella*.

We added in the sentence: “Nigratoxin was found to be lethal to both insects (Fig.1a), with an LD₅₀ of 270 ng.g⁻¹ (3 pmol.g⁻¹) for *G. mellonella* (Supplementary Figure 1) and LD₄₆ of 788 ng.g⁻¹ (9 pmol.g⁻¹) for *S. littoralis* (Fig.1a).”